# Home Dust Mites Promote MUC5AC Hyper-Expression by Modulating the sNASP/TRAF6 Axis in the Airway Epithelium

**DOI:** 10.3390/ijms23169405

**Published:** 2022-08-20

**Authors:** Ming-Zhen Chen, Shao-An Wang, Shih-Chang Hsu, Kleiton Augusto Santos Silva, Feng-Ming Yang

**Affiliations:** 1School of Respiratory Therapy, College of Medicine, Taipei Medical University, Taipei 110, Taiwan; 2Department of Emergency, School of Medicine, College of Medicine, Taipei Medical University, Taipei 110, Taiwan; 3Emergency Department, Wan Fang Hospital, Taipei 116, Taiwan; 4Department of Biomedical Sciences, Cooper Medical School of Rowan University, Camden, NJ 08103, USA

**Keywords:** house dust mites, MUC5AC, TRAF6, NASP

## Abstract

House dust mites (HDMs) are a common source of respiratory allergens responsible for allergic asthma and innate immune responses in human diseases. Since HDMs are critical factors in the triggering of allergen-induced airway mucosa from allergic asthma, we aimed to investigate the mechanisms of Toll-like receptors (TLR) in the signaling of the HDM extract that is involved in mucus hypersecretion and airway inflammation through the engagement of innate immunity. Previously, we reported that the somatic nuclear autoantigenic sperm protein (sNASP)/tumor necrosis factor receptor-associated factor 6 (TRAF6) axis controls the initiation of TLRs to maintain the homeostasis of the innate immune response. The present study showed that the HDM extract stimulated the biogenesis of Mucin 5AC (MUC5AC) in bronchial epithelial cells via the TLR2/4 signaling pathway involving MyD88 and TRAF6. Specifically, sNASP binds to TRAF6 in unstimulated bronchial epithelial cells to prevent the activation of TRAF6-depenedent kinases. Upon on HDMs’ stimulation, sNASP is phosphorylated, leading to the activation of TRAF6 downstream of the p38 MAPK and NF-κB signaling pathways. Further, NASP-knockdown enhanced TRAF6 signaling and MUC5AC biogenesis. In the HDM-induced mouse asthma model, we found that the HDM extract promoted airway hyperresponsiveness (AHR), MUC5AC, and allergen-specific IgE production as well as IL-5 and IL-13 for recruiting inflammatory cells. Treatment with the PEP-NASP peptide, a selective TRAF6-blocking peptide, ameliorated HDM-induced asthma in mice. In conclusion, this study indicated that the sNASP/TRAF6 axis plays a regulatory role in asthma by modulating mucus overproduction, and the PEP-NASP peptide might be a potential target for asthma treatment.

## 1. Introduction

Allergic asthma is a common chronic respiratory disease with clinical symptoms of wheezing, shortness of breath, and chest tightness, and it is characterized by chronic airway inflammation, airflow obstruction, airway hyper-responsiveness (AHR), and mucus hypersecretion [1]. House dust mites (HDMs) *Dermatophagoides pteronyssinus* and *Dermatophagoides farinaeis* are among the common sources of indoor allergens and the most prevalent cause of allergic sensitizations in asthma cases [2,3]. HDMs induce T helper 2 (Th2)-based immune responses, which play a prime role in orchestrating the allergic response in the airway leading to an increased production of immunoglobulin E (IgE), eosinophilia, mucus, and Th2 cytokines such as IL-4, IL-5, and IL-13 [2,4,5]. Recently, the innate immune response was recognized as an early critical mechanistic event mediating the pathogenesis of HDM-induced asthma [6,7]. HDMs contain various allergenic epitopes and pathogen-associated molecular patterns (PAMPs) that can bind to the Toll-like receptors (TLRs) that are expressed by lung epithelial cells inhabiting the airway mucosa, resulting in the production of pro-Th2 cytokines that include granulocyte macrophage colony-stimulating factor (GM-CSF), IL-25, IL-33, and thymic stromal-derived lymphopoietin (TSLP) [7,8,9]. Indeed, HDMs were shown to trigger TLR2, 4, 6, and 9 activations in the development of asthma [10,11]. However, the molecular bases of TLR signaling in the physiopathology of HDM-induced allergic asthma are still poorly defined.

Mucin 5AC (MUC5AC) is one of the major mucous gels that coat the airway’s apical surfaces and is primarily produced by epithelial goblet cells within the tracheal and bronchial epithelia [12]. Several clinical and experimental studies demonstrated that MUC5AC expression and protein production were upregulated in asthmatic patients [13,14] and that the excessive production of mucins is critical in the development of a mucus metaplasia in asthmatic airway epithelium [15]. Furthermore, elevated levels of MUC5AC contributed to the development of AHR in an asthma animal model [16,17]. It was shown that TLR-mediated myeloid differentiation primary response 88 (MyD88)/tumor necrosis factor receptor-associated factor 6 (TRAF6)-dependent signaling potentially impacts airway epithelial cell mucin production. Recent studies on airway epithelial cells demonstrated the role of TLR4 in regulating MUC5AC hypersecretion responses to bacterial products [18,19,20]. MyD88 deficiency was associated with abnormal airway epithelial MUC5AC expression and mucous cell metaplasia during TLR activation found in global MyD88-knockout (KO) mice [21]. Moreover, MyD88/TRAF6-dependent signaling was recently demonstrated to enhance the MUC5AC mucin secretion, whereas inhibition attenuated MUC5AC hypersecretion in human airway epithelial cells [18,20,22,23]. Thus, MyD88/TRAF6 signaling plays a critical role in regulating MUC5AC expression in the airway in response to TLR activation.

Somatic and testicular nuclear autoantigenic sperm protein (sNASP and tNASP, respectively), two major isoforms, were identified in humans, and both were localized in the nucleus and cytoplasm [24]. NASP was reported as a histone chaperone that regulates histone transportation, cell proliferation, and normal development [24,25,26]. In our previous studies, sNASP was found to be a critical element activating TRAF6 downstream signaling during TLR activation [27,28]. Following TLR activation, sNASP is phosphorylated and released from TRAF6 to allow downstream nuclear factor (NF)-κB signaling [28,29]. Transduced cell-permeable PEP-NASP peptides can specifically bind to TRAF6, which significantly reduced TLR4-mediated inflammation by inhibiting TRAF6 autoubiquitination and NF-κB activation [27]. However, limited information is available on the relationship between mucus hypersecretion and TLR-mediated sNASP/TRAF6 signaling in response to the effects of HDM exposure of the respiratory tract. Herein, we showed that exposure to HDM extract dramatically increased the phosphorylation of sNASP and activated TRAF6 downstream kinases in both 16HBE cells and mouse airways. As to the mechanism, the HDM extract significantly increased MUC5AC expression in human airway epithelial cells that was closely related to the TLR2/4, MAPK (p38), and NF-κB pathway activations. Moreover, targeting the sNASP/TRAF6 pathway by the PEP-NASP peptide reduced HDM-induced bronchial inflammation and MUC5AC expression. Thus, we concluded that MUC5AC expression is regulated by sNASP/TRAF6 signaling via TLR2 and TLR4, which may provide a novel therapeutic strategy for allergic asthma.

## 2. Results

### 2.1. TLR Signaling Is Required for HDM-Induced Elevation of MUC5AC Levels in 16HBE Cells

Mucus hyper-secretion is known to contribute to multiple clinical abnormalities in asthmatic patients [30]. HDMs were identified as an important factor in regulating MUC5AC [31]. In the present study, we confirmed that HDMs could induce MUC5AC mRNA and protein secretion in 16HBE cells (Figure 1A,B). There are several reports demonstrating that TLR signaling is involved in mucus secretion. To precisely identify the mechanism of HDMs in TLR signal activation, small-interfering (si)RNA targeting TLR2, -3, -4, -5, -6, -7, -8, and -9 challenged with the HDM extract was examined. We found that only the depletion of TLR2 or TLR4 significantly reduced the production of MUC5AC at the level of both the mRNA and protein in response to the HDM extract (Figure 1A,B). Furthermore, the knockdown of MyD88 and TRAF6 also dramatically inhibited the upregulation of MUC5AC induced by HDM (Figure 1A,B). The RT-PCR analysis confirmed the knock-down efficiency of siRNA (Appendix A). These results implied that the TLR2/4-MyD88–TRAF6 axis is connected to elevated MUC5AC expression in HDM-treated 16HBE cells.

### 2.2. NASP-Knockdown Enhanced MUC5AC Production by HDMs

The phosphorylation of sNASP is a critical step in the activation of innate immune responses by the TLR4/TRAF6 axis [28]. At 2 h after HDM stimulation, sNASP was serine-phosphorylated and dissociated from endogenous TRAF6 in 16HBE cells (Figure 2A). Furthermore, the down-regulation of TLR2/4 and MyD88 significantly decreased the phosphorylation of sNASP in response to HDM, which suggested that phosphorylated sNASP is involved in the HDM signaling (Appendix A). To confirm the role of sNASP in the regulation of MUC5AC production, the RNA and protein levels of MUC5AC were measured by real-time PCR, western blotting, and ELISA in siNASP-treated 16HBE cells followed by HDM exposure. The results showed that the loss of sNASP resulted in a dramatic increase in HDM-induced MUC5AC at the level of both the mRNA and protein compared to siNC (the normal control), suggesting that sNASP is required for the TLR4/TRAF6 axis to avoid eliciting MUC5AC production (Figure 2B–D). In addition, silencing NASP alone stimulated spontaneous MUC5AC production independent of the HDM stimulation (Figure 2C). To further determine whether the phosphorylation of sNASP at serine 158 regulates the interaction between TRAF6 and sNASP following an HDM stimulation, sNASP wild-type (WT) or various phosphor mutants were transfected into 16HBE cells. Two sNASP mutants were used in this study: sNASP S158A (with a serine to alanine acid substitution), a mutant serine 158 with a phosphorylation-deficient mutant residue, and sNASP S164A, which was used as a control. The results showed that the overexpression of phosphorylation-deficient sNASP S158A dramatically decreased HDM-triggered MUC5AC production compared to sNASP WT and S164A, suggesting that the non-phosphorylatable sNASP mutant S158A had a dominantly negative effect of inhibiting MUC5AC synthesis (Figure 2E).

### 2.3. The PEP-NASP Peptide Negatively Regulates sNASP/TRAF6-Meidiated NF-κB Signaling Stimulated by the HDM Extract

To further test whether sNASP is important in regulating HDM-mediated TLR signaling, NF-κB activation was examined in 16HBE cells after the HDM stimulation. As shown in Figure 3A, the HDM-induced phosphorylation of p38 MAPK and p65 was significantly increased when NASP was absent from the 16HBE cells, compared to siNC. Silencing NASP also enhanced NF-κB activity in response to HDM (Figure 3B). Since PEP-NASP, as a cell-permeable peptide, can selectively inhibit TLR4-mediated TRAF6/NF-κB signaling pathways [27], we decided to evaluate whether the transduced PEP-NASP peptide would affect TRAF6 downstream signaling following an HDM stimulation. The results indicated that the treatment of the PEP-NASP peptide, but not the control NASP peptide, diminished the HDM-induced phosphorylation of p38 MAPK and p65 (Figure 3C). The transduced PEP-NASP, but not the control NASP, was also found to inhibit HDM-mediated NF-κB activation (Figure 3D). Furthermore, the amount of MUC5AC production markedly increased upon exposure to HDM alone, but the PEP-NASP peptide attenuated the HDM-mediated MUC5AC production (Figure 3E). Collectively, these results suggested that the sNASP/TRAF6 axis is a critical regulator involved in the HDM-induced MUC5AC production.

### 2.4. The PEP-NASP Peptide Attenuates HDM-Induced Airway Inflammation

Since the sNASP/TRAF6 axis is critical for in vitro MUC5AC production, we evaluated the effect of the PEP-NASP peptide on mucus hypersecretion in vivo using an HDM-induced asthma model (Figure 4A). To determine the effect of the PEP-NASP peptide on pulmonary function, mice were exposed to methacholine aerosol. Compared to the sham control group, the HDM-treated mice had markedly higher AHR responses in terms of Newtonian resistance (Rn), total respiratory resistance (Rrs), and elastance (Ers) (Figure 4B–D). Remarkably, this upregulation of AHR was reversed by the PEP-NASP peptide treatment (Figure 4B–D). The serum HDM-specific IgE levels were significantly elevated compared to the control group, while the administration of the PEP-NASP peptide significantly decreased total IgE levels (Figure 4E).

To assess the anti-inflammatory effects of the PEP-NASP peptide, a histological analysis of lung tissues was performed using H&E and periodic-acid Schiff (PAS) staining. Our data show a robust infiltration of inflammatory cells around the airway and hyperplasia of goblet cells (PAS-positive cells) in the bronchus of HDM-treated mice compared with the control mice (Figure 5A,B and Appendix A). The treatment of the PEP-NASP peptide significantly decreased lung inflammation (Figure 5A and Appendix A) and the number of PAS-positive cells in the airways also decreased, compared to the HDM alone (Figure 5B and Appendix A). The numbers of total and differential inflammatory cells in the bronchoalveolar lavage fluid (BALF), including neutrophils, eosinophils, and macrophages, significantly increased in the HDM-treated group compared to the sham control group (Figure 5C–F). An intranasal instillation of the PEP-NASP peptide in the HDM group reduced the total number of inflammatory cells in BALF by 50%; in particular, the neutrophils, eosinophils, and macrophages were reduced by 50%, 45%, and 50%, respectively, compared to the HDM-treated mice (Figure 5D–F). Therefore, these results indicated that the HDM-induced airway inflammation was alleviated by the treatment of the PEP-NASP peptide.

### 2.5. The PEP-NASP Peptide Inhibits Th2 Inflammation and MUC5AC Expression in Mice with HDM-Induced Asthma

Next, we assessed the effects of the PEP-NASP peptide on Th2 cytokine production, since the Th2 response is a critical factor for asthma development. The results showed that type 2 cytokines (i.e., IL-5 and IL-13) in the lung tissues (Figure 6A) and BALF (Figure 6B) were increased in the HDM group compared to the control group. By contrast, the treatment of the PEP-NASP peptide resulted in significant decreases in the expressions of these cytokines (Figure 6A,B). In addition, the mRNA levels of MUC5AC were increased in the HDM-stimulated group, while PEP-NASP administration reduced the mRNA levels of MUC5AC in the lung tissues (Figure 6A). Similarly, the protein levels of MUC5AC increased with the HDM stimulation, and the PEP-NASP peptide attenuated this effect of HDM on Muc5ac secretion (Figure 6B). These results indicated that the PEP-NASP peptide inhibited HDM-induced Th2 cytokines and MUC5AC expression and secretion in this HDM-induced asthma model.

### 2.6. The PEP-NASP Peptide Inhibits the sNASP/TRAF6 Signaling in HDM-Induced Mice Asthma

To confirm whether the PEP-NASP peptide has similar effects on the regulation of *sNASP/TRAF6* and downstream NF-κB activation, as seen in previous cellular experiments, we evaluated the effects of the PEP-NASP peptide on the phosphorylation of sNASP and p65 proteins in the lung tissues of HDM-treated mice. Immunohistochemical (IHC) staining and western blotting both showed a markedly increased phosphorylation of sNASP in the alveolar areas around the airways in this HDM-induced asthma model (Figure 7A,B). Treatments of the PEP-NASP peptide caused significantly lower sNASP phosphorylation (Figure 7A,B). The western blot analysis revealed that the HDM induced the phosphorylation of p65 in lung tissues (Figure 7B,C). These alterations in the phosphorylation of p65 were reversed by the PEP-NASP peptide (Figure 7B,C). Moreover, the protein levels of MUC5AC were increased by HDM exposure in the lungs, and the PEP-NASP peptide attenuated this effect of the HDM on the Muc5ac expression levels (Figure 7B,C). Taken together, these data suggested that the PEP-NASP peptide inhibited airway inflammation and mucin production by suppressing NF-κB signaling in this HDM-induced asthma mouse model.

## 3. Discussion

Allergic asthma is a complex, chronic, and heterogeneous airway disorder with different phenotypes caused by various factors, and it has imposed a substantial economic burden on healthcare systems in low and middle-income countries, causing approximately 250,000 deaths annually [32,33]. The current knowledge about cell-signaling pathways, especially within cells that have central roles in the pathophysiology of asthma, has shown promise in providing new hope for developing more-effective anti-asthmatic agents [34]. Thus, we investigated the mechanism of HDM-induced MUC5AC production—which is a major functional phenotype of airway epithelial cells—with respect to TLR signaling pathways. Our study found that exposure to HDM induced both MUC5AC mRNA and protein expressions via the TLR2/4-mediated sNASP/TRAF6 pathway in an in vitro model. After the HDM stimulation, sNASP was phosphorylated at serine 158 and was dissociated from TRAF6 to allow downstream NF-κB signaling. Moreover, increased phosphorylation levels of sNASP were detected in both the in vitro cellular model and the in vivo HDM-induced asthma animal model. The results obtained in the present study also demonstrated the effectiveness of the PEP-NASP peptide, which targets sNASP/TRAF6 signaling, in reducing of the inflammatory cell profile, type two cytokines, AHR, and airway mucus in an experimental model of chronic allergic lung disease induced by HDM. To our knowledge, this is a seminal study demonstrating the role of sNASP/TRAF6 signaling in mediating mucin hyperexpression associated with HDM-induced asthma.

Since mucus on the respiratory surface represents the first line of defense for preventing the penetration of pathogens and foreign particulates, the hypersecretion and accumulation of mucus and changes in its biochemical/biophysical properties can become major contributors to respiratory diseases, such as asthma and chronic obstructive pulmonary disease (COPD) [35]. Of the numerous mucin (the glycoprotein component of mucus) genes detected in the human respiratory tract, only *MUC5AC* and *MUC5B* have established roles in health and disease [36]. MUC5B is crucial for the innate defense of the lungs’ and predominates in healthy human lungs; its levels in asthma either remain stable or decrease [15,36,37]. On the other hand, MUC5AC levels in asthma disproportionately and consistently increase in response to cigarette smoke and allergens [15,36,37,38]. In patients with type 2-high asthma, an increased production of MUC5AC is correlated with mucus hypersecretion, airflow obstruction, and airway hyper-responsiveness [39]. The functional significance of MUC5AC in allergic AHR, inflammation, and mucus obstruction was also proven in *Muc5ac^−/−^* mice [16]. Therefore, we focused on the overexpression of the *MUC5AC* genes and the related signaling pathways induced by HDM extracts in bronchial epithelial cells and an HDM-induced asthma animal model. Our study demonstrated that HDM induced the protein expression of MUC5AC, but not MUC5B (data not shown), through the activation of the TLR2/4, TRAF6, MAPK (p38), and NF-κB signaling pathways.

MUS5AC has been considered as a central effector of allergic inflammation, which is required for AHR [40]. However, the link between MUC5AC regulation and inflammation is incompletely understood. A recent study found that eosinophils in ovalbumin- and *Aspergillus oryzae* extract-challenged mice did not produce any observable difference between WT and Muc5ac^−/−^ lungs. By contrast, AHR in the Muc5ac^−/−^ lung were significantly reduced, suggesting that MUC5AC is essential for allergic AHR but may not be necessary for allergic inflammation [16]. Others also demonstrated that complement C3a anaphylatoxin and its receptor regulates mucin production in T-cell-deficient mice and in the absence of IL-13 [41]. The data here showed that the sNASP/TRAF6 axis regulates MUC5AC production and type two inflammation; however, it cannot rule out an additive impact on mucin hyper-expression by airway epithelial cells independently of type two responses. It is possible that MUC5AC directly contributes to mucous hyper-expression and causes AHR in asthmatic patients, since the protection from AHR was not mediated by altered inflammation but through reduced mucin formation [16]. Thus, it remains to be determined whether the mechanism by which the sNASP/TRAF6 axis promotes mucin production also directly crosstalks to type two inflammation in asthmatic cells.

TLR recognize antigens through pathogen-associated molecular patterns (PAMPs) such as bacterial lipopolysaccharide (LPS) or damage-associated molecular patterns (DAMPs) such as heat-shock proteins or high-mobility group box 1 [42]. Genetic polymorphisms involved in TLR signaling pathways have been related to the development of asthma [43,44]. Moreover, several studies revealed that there are positive correlations of TLR2 and TLR4 with allergic asthma [45,46]. HDMs and their major allergen, *Dermatophagoides pteronyssinus*, were reported to act in both TLR2 and TLR4 signaling [47,48]. Both TLR2 and TLR4 have different signaling pathways but share the MyD88 adaptor molecule. In previous studies, MUC5AC expression and secretion were closely associated with the TLR4/MyD88-mediated signaling pathways. The inactivation of the TLR4/MyD88 pathway by small molecules or siRNA significantly attenuated MUC5AC expression [18,19]. However, enhanced MUC5AC levels were found in global MyD88-KO mice with TLR activation by agricultural organic dust extract (ODE) [21], suggesting that the role of TLR4/MyD88 pathway in the regulation of MUC5AC expression is still controversial. Our results supported that TLR4/MyD88 signaling is a positive regulator of HDM-induced MUC5AC production. Silencing TLR2, TLR4, or MyD88 inhibited HDM-induced MUC5AC synthesis and secretion, as shown in the airway epithelial cells. Furthermore, the inhibition of TLR4/MyD88 signaling by the PEP-NASP peptide in our HDM-induced asthma model significantly decreased mucin hyperexpression and airway inflammation. Thus, our findings may provide insights into the function of TLR/MyD88 signaling in mucin secretion.

TRAF6 is a major signaling node in TLR signaling for maintaining immune homeostasis since the hyperactivation or hypoactivation of TLR signaling can lead to various human diseases [49]. Our previous work showed that the sNASP is a negative regulator of TLR/TRAF6-dependent NF-κB signaling and may contribute to regulating airway innate immunity and inflammation [28,29]. A couple of reports have recently demonstrated that the upregulation of MUC5AC in airway epithelial cells was regulated by TLR4-dependent MyD88-TRAF6-ERK signaling via PKCθ, MKP1, and IκB Kinase, suggesting that ERK activation appears to be important in mediating MUC5AC production [22,23,50]. While using a miR-146a mimic inhibited the expressions of proteins relevant to the TLR4/TRAF6/NF-κB signaling pathway, it also inhibited the stimulation of MUC5A in 16HBE cells [51]. Thus, in the present study, we explored the functional role of the sNASP in TLR/TRAF6-dependent NF-κB signaling and provided the mechanism to support the previous discovery. In unstimulated bronchial epithelial cells, the sNASP binds to TRAF6 to avoid excessive mucin production and overwhelming the mucociliary clearance mechanism, which may lead to defective mucosal defenses. Once the HDMs activated TLR2 or TLR4 signaling, sNASP is phosphorylated and released from TRAF6, which ultimately stimulates MUC5AC expression. Thus, a precise regulation of sNASP/TRAF6 signaling is critical for maintaining an appropriate mucin balance between beneficial and detrimental outcomes.

Since mucus accumulation is a major cause of airway obstruction in asthma, mucoactive agents have been used to modify mucus production and facilitate mucus clearance or inhibit mucus secretion [52,53]. For instance, the synergism of inhaled corticosteroid and β-adrenergic agonist treatments of human bronchial epithelial cells inhibited the production of pathologic mucins through decreasing IL-13-induced MUC5AC-positive goblet cell metaplasia and mucus secretion [17]. Recently, other approaches to regulating mucus obstruction have been proposed to include the specific targeting of the MUC5AC gene transcription or protein processing or blocking the differentiation of mucus-producing goblet cells. For example, targeting Notch or TLR signaling relieved goblet cell metaplasia and mucin production, and these can act as novel agents against allergic asthma [54,55,56]. Prunetin and euxanthone extracted from an herbal route can both attenuate TLR4/MyD88-mediated airway inflammatory responses and MUC5AC expression, suggesting that TLR/MyD88 signaling pathways may be important targets for treating diseases. In this study, the blockade of TLR4/TARF6 signaling by the cell permeable PEP-NASP peptide was used to demonstrate that the targeting of sNASP/TRAF6 can be an important mechanism for treating diseases. In our HDM-induced asthma model, we found that PEP-NASP peptide treatment markedly ameliorated AHR and reduced the production of type two inflammatory cytokines (IL-5 and IL-13) and MUC5AC. The mechanisms for the protective role of the PEP-NASP peptide are associated with the inhibition of TLR4-mediated TRAF6/NF-κB signaling pathways. Overall, these results indicated that targeting the sNASP/TRAF6 axis in asthma might be a protective mechanism partly by reducing mucin overproduction in asthmatic airways.

## 4. Materials and Methods

### 4.1. Cells and Reagents

Human 16HBE bronchial epithelial cells were kindly provided by Shih-Chang Hsu (Wan Fang Hospital, Taipei, Taiwan). The 16HBE cells were maintained in DMEM medium (ThermoFisher Scientific, Carlsbad, CA, USA) supplemented with 10% fetal bovine serum (Corning, Corning, NY, USA), 100 U/mL penicillin and 0.1 mg/mL streptomycin (ThermoFisher Scientific, Carlsbad, CA, USA) and were cultured in a humidified chamber containing 5% CO_2_ at 37 °C.

The HDM extract (*D. pteronyssinus/ D. farinae extract* mix) was purchased from Citeq (#20.50.85, Groningen, The Netherlands). Preparation of the PEP-NASP and NASP peptides was previously described in [27]. PEP-NASP or NASP peptides (0.4 μM) were added to the culture medium for 1 h before HDM treatment. The following monoclonal antibodies were used in this study: anti-NASP (SC-161915, Santa Cruz, Dallas, TX, USA), mouse anti-phosphoserine (ab17465, Abcam, Cambridge, UK), rabbit anti-TRAF6 (SC-7221, Santa Cruz), mouse MUC5AC (SC-21701, Santa Cruz), mouse anti-GFP (632281, Clontech, San Jose, CA, USA), mouse anti-His (SC-8036, Santa Cruz), anti–phosphop38MAPK (9211, Cell Signaling, Danvers, MX, USA), anti-p38 MAPK (9212, Cell Signaling), mouse anti-phospho-NF-κB p65 (GTX54672, Genetex, Hsinchu, Taiwan), mouse anti- NF-κB p65 (SC-372, Santa Cruz), mouse anti-β-actin (A1544, Sigma-Aldrich, St. Louis, MO, USA), rabbit anti-mouse IgG H&L (HRP) conjugated antibody (GTX213111-01, Genetex), and goat anti-rabbit IgG H&L (HRP) conjugated antibody (GTX213110-01, Genetex). Antibody dilutions were employed according to the manufacturers’ instructions. Anti-pSerine158-sNASP was generated by phage library in our previous work [28].

### 4.2. HDM-Induced Asthma Model

Animal protocols were approved by the animal and ethics review committee of the Laboratory Animal Center at Taipei Medical University. BALB/c mice at 8~10 weeks old were obtained from the National Laboratory Animal Center (Taipei, Taiwan) and housed under pathogen-free conditions for 1 week before the start of the experiments.

Animals were randomly divided into four experimental groups (10 mice per group): sham + NASP, sham + PEP-NASP, HDM + NASP, and HDM + PEP-NASP groups. The HDM-induced asthma model was established according to previous publication [57]. Briefly, the 25 μg of the HDM extract was re-suspended in 35 μL of saline and mice were sensitized by intranasal (i.n.) instillation for 5 consecutive days weekly (from Monday to Friday) for 4 weeks. Intranasal instillation was performed as described previously [58].

Sham control mice models were only treated with sterile saline and handled in a similar procedure. The PEP-NASP and NASP peptides (0.1 mg/kg) were dissolved in 0.1% DMSO and injected by intranasal 1 h before HDM challenge on Monday and Thursday from weeks 2 to 4. Twenty-four hours after the last HDM or saline exposure, airway responsiveness was measured by Flexivent (SCIREQ, Tempe, AZ, USA), as described below. After completion of the AHR measurement, mice were disconnected from the ventilator and bronchoalveolar lavage (BAL) was immediately collected. Finally, blood, lung sections and samples were collected. For the histopathological analysis, lung sections were stained with hematoxylin and eosin (H&E, Abcam) and periodic acid Schiff (PAS, Abcam), according to the manufacturer’s protocols. Phosphorylation of sNASP in the lung sections was detected by using anti–phospho-sNASP S158. Stained slides were visualized, and images were captured with an Echo Revolve Fluorescence Microscope.

### 4.3. Bronchoalveolar Lavage Fluid (BALF)

BALF was harvested according to a previous publication [27]. Briefly, BALF was obtained by injecting ice-cold PBS using a tracheal catheter. The total number of inflammatory cells, neutrophils, eosinophils, and macrophages were counted with a hematocytometer and Wright–Giemsa staining (K1438-30, Biovision, Milpitas, CA, USA). Following staining, 500 cells are counted under a microscope and manually classified into neutrophils, eosinophils, macrophages, basophils, etc.

### 4.4. Measurement of AHR

Twenty-four hours after the last HDM or saline challenge, the AHR was assessed by Flexivent system (SCIREQ) according to the manufacturer’s protocol as previously described [59]. Briefly, mice were anesthetized by ketamine (100 mg/kg IP) and xylazine (10 mg/kg IP), and their tracheas were then surgically exposed and connected to the ventilator. Mice received methacholine using a nebulizer and were sequentially exposed to increasing doses of methacholine in in phosphate-buffered saline (PBS) at 0, 12.5, 25, and 50 mg/mL (Sigma-Aldrich Inc.)

### 4.5. Plasmids, siRNA, and Transfection

The green fluorescent protein (GFP)-sNASP WT and mutants’ (S158A and S164A) plasmids were previously described [28]. The 16HBE cells were transfected using Lipofectamine 3000 (L3000015, Invitrogen, Waltham, MA, USA) according to the manufacturer’s protocol.

The siRNAs were transfected by using the DharmaFECT 1 reagent (T-2001-02, Dharmacon, Cambridge, UK) according to the manufacturer’s protocol. Sequences for siRNA are listed as follows: the siRNA specific for the gene encoding human NASP (siNASP, 5′-GGAACUGCUACCCGAAAUU-3′), TLR2 (siTLR2, 5′-GGCAGUCUUGAACAUUUAGACUUAU-3′), TLR4 (siTLR4, 5′-GGGCAUGCCUGUGCUGAGUUUGAAU-3′), MyD88 (siMyD88, 5′-AUCGCGGUCAGACACACACAACUUC-3′), and TRAF6 (siTRAF6, 5′-CCACGAAGAGAUAAUGGAUdTdT-3′). All siRNAs were synthesized by GENOMICS (Taipei, Taiwan). An siRNA Universal Negative control was purchased from Sigma-Aldrich (catalog S1C001).

### 4.6. Real-Time Polymerase Chain Reaction (RT-PCR)

Total RNA was extracted from 16HBE cell pellets or homogenized mouse lung tissues. For RNA isolation, the Toolsmart RNA extractor (TB-DPT-BD24, Biotools, Taipei, Taiwan) was used according to the manufacturer’s instructions. Complementary (c)DNA was synthesized from 1 μg of total RNA using the magic RT cDNA synthesis kit (Bio-Genesis Technologies, Taipei, Taiwan). A real-time PCR was performed using SYBR Green Master Mix (ThermoFisher Scientific) on an StepOnePlusTM machine (ThermoFisher Scientific). Relative gene expression levels were normalized to the expression of the gene encoding 18s and computed using the 2^−ΔΔCT^ method. The following PCR primers were used: human MUC5AC primers, forward—5′-AGGCCAGCTACCGGGCCGGCCAGACCA-3′, reverse—5′-GTCCCCGTACACGGCGCAGGTGGCCAG-3′; mouse Muc5ac primers, forward—5′-TACCACTCCCTGCTTCTGCAGCGTGTCA-3′, reverse—5′-ATAGTAACAGTGGCCATCAAGGTCTGTCT-3′; human IL-5 primers, forward—5′-TCAAACTGTCCGTGGGGGTA-3′, reverse—5′-AATCCAGGAACTGCCTCGTC-3′; human IL-13 primers, forward—5′-CAGCTCCCTGGTTCTCTCAC-3′, reverse—5′-CAGGGGAGTCTGGTCTTGTG-3′; human TLR2 primers forward—5′-GGCCAGCAAATTACCTGTGT-3′, reverse—5′-AGGCGGACATCCTGAACCT-3′; human TLR4 primers, forward—5′-CTGCAATGGATCAAGGACCA-3′, reverse—5′-TTATCTGAAGGTGTTGCACATTCC-3′; human MyD88 primers, forward—5′-GCAAAGAACATGGTCCGATA-3′, reverse—5′-CGTCAGTCTGTAGGTATG-3′; human TRAF6 primers, forward—5′-GCAGTGAGCCACCAGCATTT-3′, reverse—5′-AGCGACCCTGTACCAAGTT-3′.

### 4.7. Co-Immunoprecipitation (IP) and Western Blot Analysis

IP and Western blot analyses were performed as previously described [29]. Briefly, cells were lysed by RIPA lysis buffer (RIP001, Bioman, Taipei, Taiwan) supplemented with protease inhibitors (P8340, Sigma). For co-IP, whole-cell extracts obtained by centrifugation were incubated with 1 μg of the indicated antibody overnight at 4 °C and subsequently incubated with protein G Sepharose beads (16-266, Millipore, Burlington, MA, USA) for 1 h at 4 °C. After being washed twice with lysis buffer, the immunoprecipitated products were eluted using 2× protein sample buffer and boiled at 100 °C for 5 min. Protein samples were then subjected to 10% SDS-PAGE gel and transferred onto PVDF membrane for immunoblotting. Immunoblotting was performed following standard procedures. Finally, the images of immunobloting were visualized using UVP BioSpectrum 815 imaging System (Jena, Germany) and images were analyzed using ImageJ image analytical software.

### 4.8. Enzyme-Linked Immunosorbent Assay (ELISA)

The collected BALF, lung tissue, or serum were assessed for expressions of IL-5, IL-13, and IgE levels using ELISA MAX Deluxe sets (Biolegend, San Diego, CA, USA) and a MUC5AC ELISA kit (Elabscience, Houston, TX, USA), according to the manufacturer’s instructions.

### 4.9. NF-κB Reporter Assay

NF-κB-dependent luciferase activities were measured as previously described [60]. Briefly, 16HBE cells were seeded in 24-well plates 1 day before transfection. At 24 h after transfection with the indicated plasmid or siRNA, cells were stimulated with the HDM extract for another 24 h. In addition, the NASP or PEP-NASP peptides were administrated 1 h before HDM stimulation. Then cell lysates were harvested and followed by measuring the fluorescence using the Dual-Glo Luciferase Assay System (E2920, Promega, Madison, WI, USA).

### 4.10. Statistical Analysis

All statistical analyses were performed using GraphPad Prism 8 software (GraphPad, La Jolla, CA, USA) and values are presented as the mean ± standard error of the mean (SEM). A one-way analysis of variance (ANOVA) was used to compare differences between three or more groups. Student’s *t*-test was used to compare the statistical difference between two groups. * *p* < 0.05 and ** *p* < 0.01 were considered statistically significant.

## Figures and Tables

**Figure 1 ijms-23-09405-f001:**
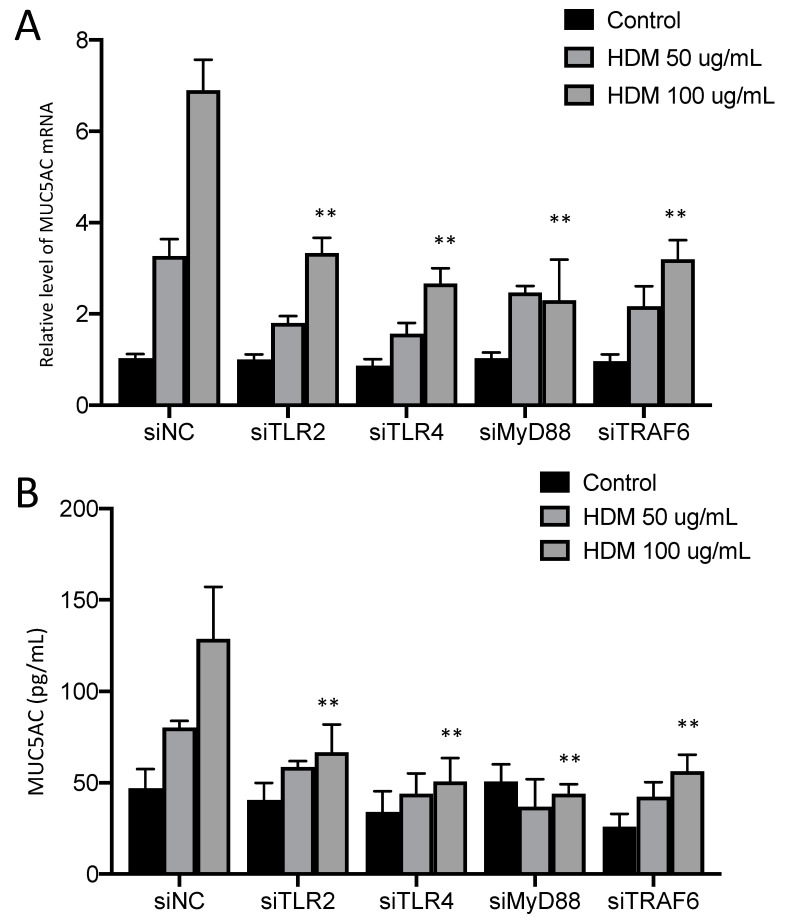
HDM induces MUC5AC production in airway epithelial cells via TLR2/4-MyD88 signaling. (**A**) Expression of MUC5AC in 16HBE cells transfected with indicated siRNA and stimulated with HDM for 24 h. Results were normalized to the expression of ACTB (encoding β-actin) and control (siNC) cells. (**B**) Secretion of MUC5AC by 16HBE cells transduced as in (**A**) and stimulated with HDM. Data are representative of three independent experiments and presented as the mean ± SEM. ** *p* < 0.01.

**Figure 2 ijms-23-09405-f002:**
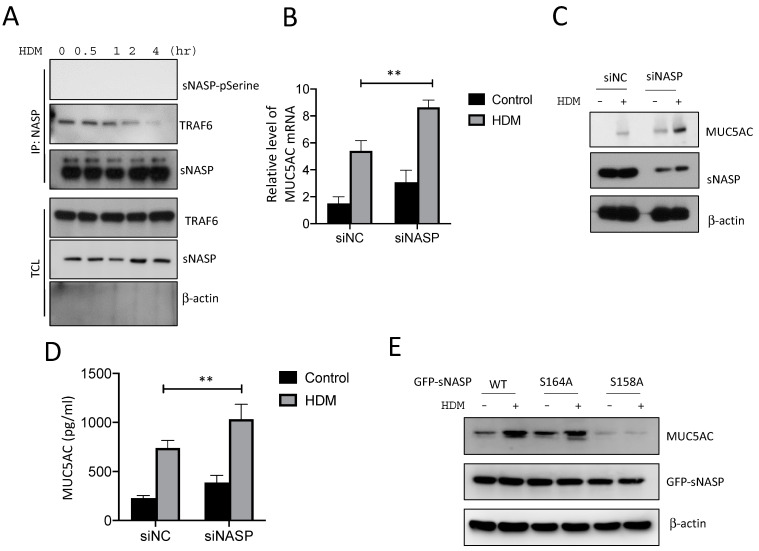
Phosphorylation of sNASP regulates its interaction with TRAF6 and affects MUC5AC production following exposure to HDM. (**A**) The 16HBE cells were treated with HDM for different time periods and assessed by immunoblotting (IB) with antibodies against phosphorylated serine, NASP, or TRAF6 after immunoprecipitation (IP) with anti-NASP or by IB with anti-TRAF6 and anti-β-actin in total cell lysate (TCL). (**B**) RNA levels of MUC5AC in 16HBE cells transfected with the indicated siRNA and stimulated with HDM for 24 h. Results were normalized to the expression of ACTB (encoding β-actin) and control (siNC) cells. Data are presented as the mean ± SEM. ** *p* < 0.01 (**C**) Protein levels and (**D**) secretion of MUC5AC by 16HBE cells transduced as in (**B**) and stimulated with HDM. (**E**) The 16HBE cells were transfected with GFP-tagged wild-type (WT) sNASP or S158A and S164A mutants, followed by IB with antibody MUC5AC, GFP, or β-actin. Data are representative of three independent experiments.

**Figure 3 ijms-23-09405-f003:**
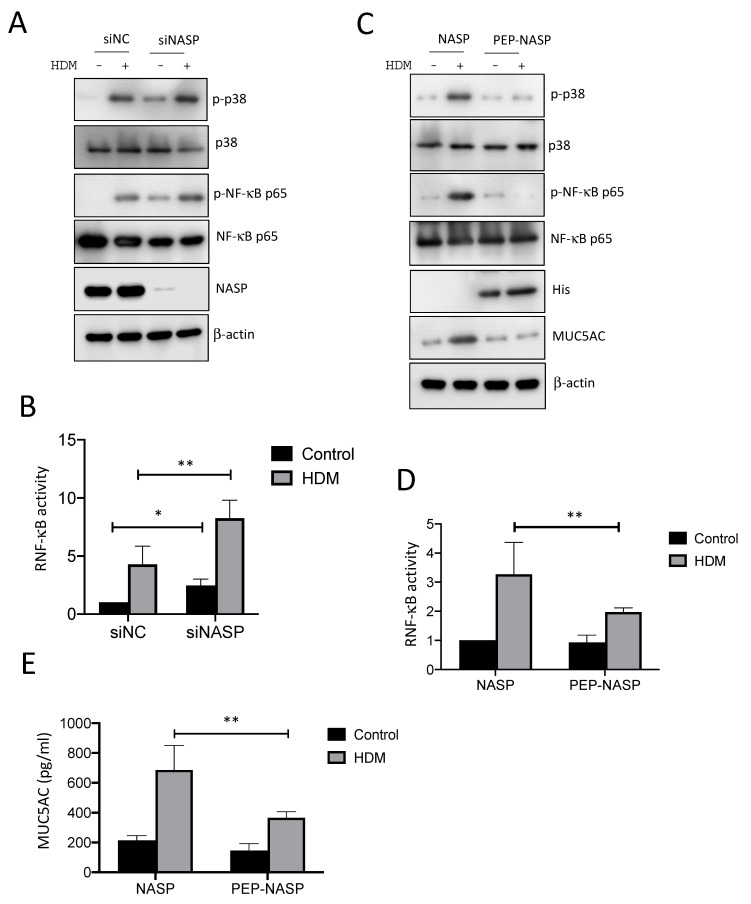
The PEP-NASP peptide negatively regulates MUC5AC via NF-κB signaling. (**A**) The 16HBE cells were transfected with the siNC or siNASP, and then stimulated with HDM and assessed by an immunoblotting (IB) analysis with the indicated antibodies. (**B**) Luciferase activity in 16HBE cells transfected with a luciferase reporter vector driven by an NF-κB-responsive promoter, plus the siNC or NASP, and stimulated with HDM. Results were standardized to a control (siNC; set to 1). Data are the mean ± SEM for each group. (**C**) The 16HBE cells pretreated with the PEP-NASP or NASP were stimulated with HDM for and assessed by IB analysis with the indicated antibodies. (**D**) Luciferase activity of NF-κB (**E**) Secretion of MUC5AC by 16HBE cells transduced as described in (**C**). Data are representative of three independent experiments. * *p* < 0.05 and ** *p* < 0.01 (by a one-way ANOVA).

**Figure 4 ijms-23-09405-f004:**
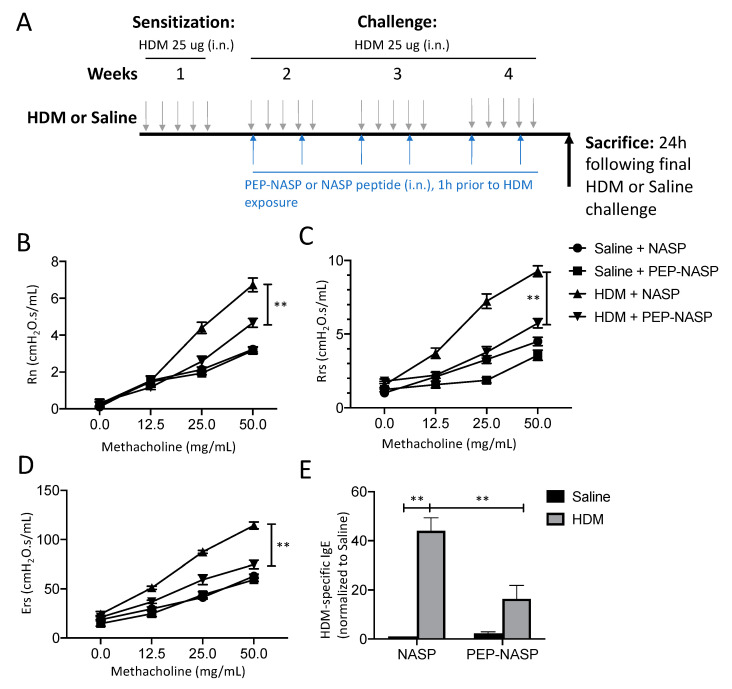
Effect of the PEP-NASP peptide on airway hyperresponsiveness in HDM-induced asthma model. (**A**) Timelines of HDM-induced airway inflammation model. Mice were sensitized to 25 μg of HDM in 35 μL saline or 35 μL saline alone by intranasal instillation for 5 days/week for 4 weeks. The PEP-NASP or NASP peptide was treated 1 h before HDM stimulation on days 1 and Day 3 from weeks 2 to Week 4. (**B**) Newtonian resistance (Rn), (**C**) resistance (Rrs), (**D**) elastance, and (**E**) serum HDM-specific IgE levels were assessed from each group. Results are mean values, with error bars representing SEM. (** *p* < 0.001, compared with saline-treated control mice).

**Figure 5 ijms-23-09405-f005:**
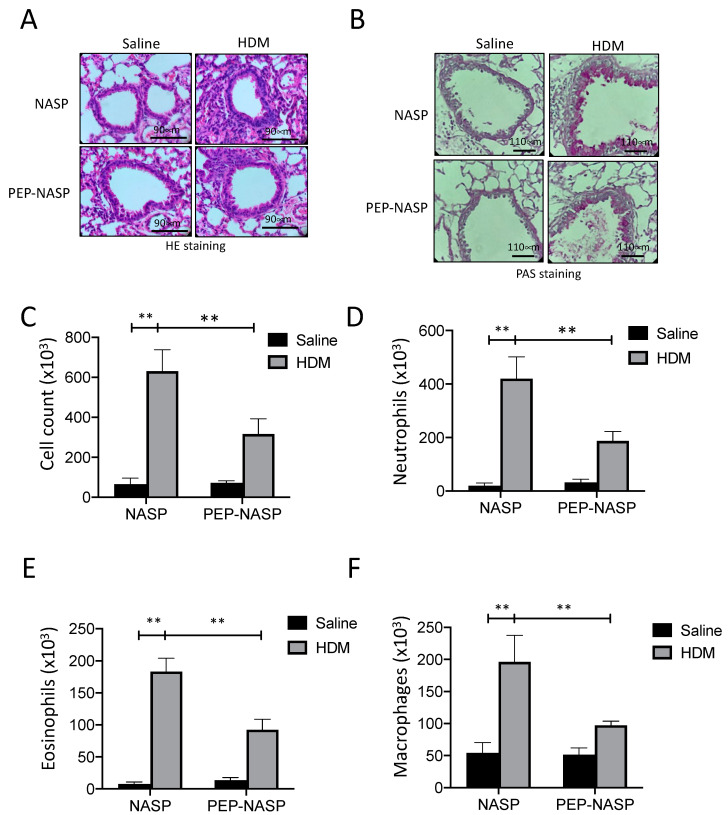
Effects of the PEP-NASP peptide on airway inflammation and mucus secretion in HDM-induced asthma model. Lungs from each group of mice were fixed and stained with H&E (**A**) and PAS (**B**). Images of lung sections were captured by microscope (magnification, ×400); scale bar = 90 μm and 110 μm. (**C**) Total BALF cell counts (2 × 10^6^) were determined by a hemocytometer. (**D**) Neutrophil, (**E**) eosinophil, and (**F**) macrophage counts were determined in 500 total BALF cells after Wright–Giemsa staining. ** *p* < 0.01 (by a one-way ANOVA). (*n* = 10 per group per experiment).

**Figure 6 ijms-23-09405-f006:**
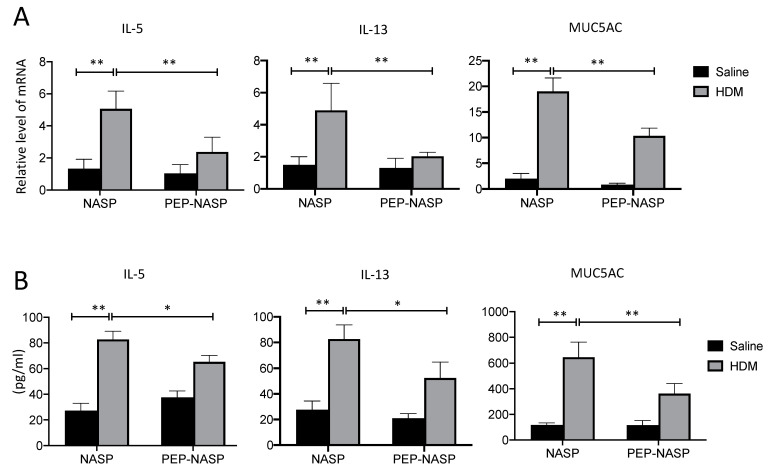
Effects of the PEP-NASP peptide on mRNA expression and secretion of inflammatory cytokines in BALF and lung tissues. (**A**) RNA expressions of IL-5, IL-13, and MUC5AC in lung tissues were determined by a RT-qPCR. (**B**) Levels of IL-5, IL-13, and MUC5AC in bronchoalveolar lavage fluid (BALF) were detected by an ELISA. *n* = 10, * *p* < 0.05 and ** *p* < 0.01 (by a one-way ANOVA).

**Figure 7 ijms-23-09405-f007:**
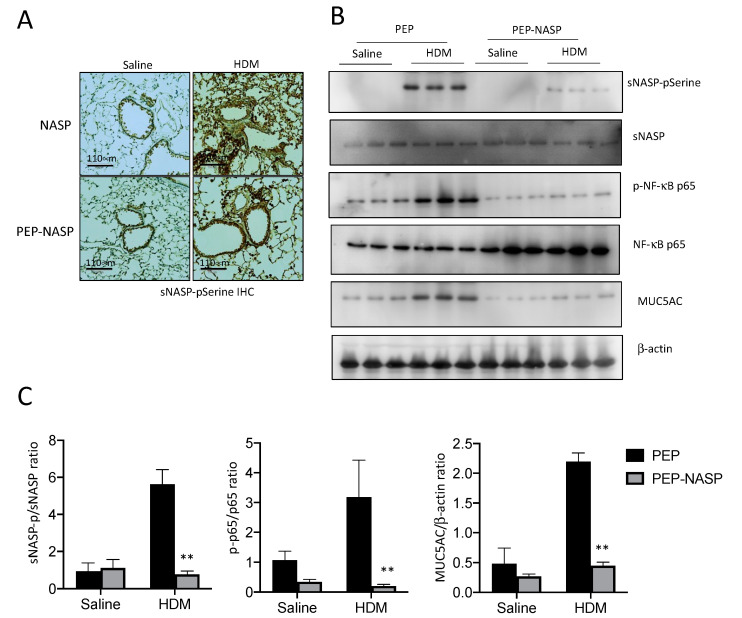
Effects of the PEP-NASP peptide on the phosphorylation of sNASP in lung tissues. (**A**) Phospho-NASP-stained lung sections from each group were highlighted using immunohistochemical (IHC) staining by anti-pSerine158-sNASP. Scale bar = 100 μm (**B**) Lung from each group was analyzed by Western blotting with pSerine158-sNASP, sNASP, p-NF-κB p65, NF-κB p65, MUC5AC and β-actin antibodies. (**C**) Densitometric analysis of the Western blot shown below. Data represent the mean ± SEM of eight to twelve experiments (** *p*  <  0.001 compared to PEP-treated control mice).

## Data Availability

All data are reported in the present study.

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
