# Peer review of "Home Dust Mites Promote MUC5AC Hyper-Expression by Modulating the sNASP/TRAF6 Axis in the Airway Epithelium"

_ijms, 2022, doi:10.3390/ijms23169405_

Round 1
Reviewer 1 Report
The manuscript by Chen et al. entitled: “Home dust mites promote MUC5AC hyper-expression by modulating the sNASP/TRAF6 axis in the airway epithelium” presents a study on the contribution of sNASP to the HDM-induced MUC5A expression and allergic asthma through modulating TLR2/4 and MyD88-dependent TRAF6 activation pathway. The scientific progress in this study is a bear minimal from their previous works. The manuscript is relatively well organized. The result of this study provides interesting insights to use cell permeable PEP-NASP peptides for the treatment of HDM-induced allergic asthma.
The following are some minor concerns that require clarification:
1. Define abbreviation when it first used [e.g., somatic nuclear autoantigenic sperm protein (sNASP)].
2. May be need to do proofreading and editing by a native English-speaking scientific writer/editor.
3. Provide little bit more information on NASP function, variant forms, cell distribution, etc. in addition to relation to TLR signaling.
4. Provide background information about the PEP-NASP, especially in context with TRAF6 interactions.
5. Introduction (page 2), define “high TLRs activation”.
6. Lack of PEP or PEP-scramble control treatment.
7. Page 2 (and throughout the manuscript), “TLRs signaling” should be “TLR signaling”.
8. Fig. 1, need siRNA-knockdown specificity and efficacy data need to be provided.
9. Fig. 2A, This reviewer don’t see any phosphorylated sNASP bands or actin bands.
10. Fig. 2A, also need input (TCL) sNASP blot.
11. Page 3, “…in the absence and presence of siRNA NASP (siNASP)” should be “in the absence or presence of siRNA NASP (siNASP)”
12. Page 3, “To further confirm how sNASP func-tions in regulating of MUC5AC production, we performed real-time PCR, western blot-ting and ELISA in the absence and presence of siRNA NASP (siNASP) and stimulated these cells with the HDM extract.” Clarify this sentence.
13. Fig. 3, define RNF-kB activity.
14. Page 6, “Remarkably, ….. by the transduction of the PEP-NASP peptide (Figure 4B-D).” change to “… by the PEP-NASP peptide treatments….”
15. Fig. 4E, fix X-axis label.
16. Fig. 4E, clarify whether it is serum IgE levels or BALF IgE levels. In figure 4 legend, it expressed as serum IgE but, in the Method section, it looks like measured in BALF.
17. Fig. 5A and 5B, provide low magnification slides too (either in Fig. 5 or as a supplemental materials).
18. Fig. 5A & 5B, put scale bar in all tissue section slides.
19. Fig. 5A & 5B, check the unit symbol in the scale bar. It looks like “¥m” on my computer screen.
20. Fig. 5C, 5D, & 5F, fix X-axis label.
21. Fig. 5A scale bar is 90 mm while Fig. 5B scale bar is 110 mm in the figure slide. Clarify in the figure legend.
22. Fig. 5C-5F, in the figure they are presented as cell # but figure legend said they are percentage. You may want to clarify it by explaining what you did a little bit more detail.
23. Fig. 5C-5F, provide how many total cells were counted to get % of each type of cells after Wright-Giemsa staining.
24. Fig. 5C-5F, provide (either in Fig. 5 or in supplemental data) representative Wright-Giemsa staining slide of BAL cells with indication of each cell type.
25. Page 8 last paragraph, “….we evaluated effects of the PEP-NASP peptide on the phosphorylation of sNASP and protein expression of p65 proteins …” should be “……on the phosphorylation of sNASP and p65 proteins…”. You did evaluate phosphorylation status of p65 in lung tissue not protein expression of p65.
26. Page 8 last line, “Transduction of the PEP-NASP..” change to “Treatments of …”.
27. Page 9 first sentence, “..lower expression of sNASP phosphorylation (Figure 7A and B)” should be “…lower sNASP phosphorylation (Figure 7A and B)”.
28. Fig. 7A, scale unit is not indicated.
29. Fig. 7, when animal is harvested? 24 hr after the last HDM challenge? How long sNASP and p65 remain phosphorylated after HDM stimulation?
30. Page 10, What is “high TLR”?
31. Provide evidence that HDM induced phosphorylation of sNASP is inhibited in TLR2siRNA-, TLR4siRNA-, or MyD88siRNA-transfected cells.
32. Page 11, PEP-sNASP peptide vs PEP-NASP peptide. I assume the authors indicate the same peptide. Please be consistent.
33. Information on concentrations of PEP-NASP and NASP peptides used for in vivo and in vitro experiments is not provided.
34. Information on PEP-NASP and NASP peptide administration rout is not provided.
35. Provide siNC sequence information.
36. Clarify the sentence “The immunoprecipitated beads were then using 2× Laemmli sample buffer and separated by SDS- PAGE.”
37. Clarify the sentence “Finally, the PVDF membranes were visualized using UVP BioSpectrum 815 (Jena, Germany) and an-alyzed using ImageJ image analytical software.”
38. Description on Immunohistochemistry is missing in the Materials and Methods section.
39. Description on lung tissue histology is missing in the Materials and Methods section.
Author Response
Thanks for the reviewer’s comments concerning our manuscript titled “Home dust mites promote MUC5AC hyper-expression by modulating the sNASP/TRAF6 axis in the airway epithelium” (ID: ijms-1801706). Your comments are all valuable and very helpful for revising and improving our paper. We have studied comments carefully and have made correction which we hope will meet with approval. Please see the attachment for the point-by point response.

Reviewer 2 Report
General comments
In the present manuscript, Chen et al. report that the sNASP/TRAF6 axis regulates mucus production and airway inflammation by using a bronchial epithelial cell line and a house dust mite asthma model.
The research question is of interest as there is currently limited data on the role of the sNASP/TRAF6 axis in allergic asthma. The experiments are well conducted.
Major
- The manuscript has English language issues that needs revision. Moreover, the background is somewhat approximative and not always up to date in the context of allergic asthma. Many terms are not appropriate for example: HDM-induced lung injury (I suggest to replace this by HDM induced bronchial inflammation …)
Minor
- Can the authors more deeply discuss the link between MUC5AC regulation and airway inflammation in the context of asthma as both are not necessarly associated
Author Response

(The authors gave the same response as above.)

Round 2
Reviewer 1 Report
Major concerns must be addressed:
1. Fig. 2A, This reviewer see neither phosphorylated sNASP bands nor actin bands. Check these blots presented in Fig. 2A and provide correct blots.
2. Page 3 &4, “At 30 minutes after HDM stimulation, sNASP was serine-phosphorylated in 16HBE cells (Figure 2A). Similar to our previous finding, endog-enous TRAF6 dissociation from sNASP was correlated with increased phosphorylation of sNASP in response to HDM (Figure 2A). These results suggested that HDM-treated bron-chial epithelial cells can initiate phosphorylation of sNASP, which leads to the release of TRAF6 from sNASP to allow downstream signaling”. These claim by the authors are not supported by the data presented in Fig. 2A. Either show the supporting data or delete this claim.
3. Figure 7, The authors must provide information on how they detect the phosphorylated form of sNASP in IHC and WB and how they validate what the presented data.
4. Page 13 “Phosphorylation of sNASP in the lung sections were detected by using anti–phospho-sNASP S158.” Provide source of anti-phospho-sNASP S158. If it is a custom antibody and reported previously, please cite the previous paper that showing validation of the antibody. If it has not been previously reported, please describe how the antibody is generated and validated.
Minor concerns need to be clarified:
1. Page 2 and throughout the manuscript, “… TLRs activation” should be “.. TLR activation”. or “…. activation of TLRs”.
2. Page 2 and throughout the manuscript, “…TLRs-mediated..” & “…TLRs signaling..” should be “.. TLR-mediated..” & “..TLR signaling..”, respectively.
3. Page 3 & page 4, “..at the level of both the mRNA and protein levels ….” remove “levels” following protein.
4. Supplementary information, it would be better if the authors provide evidence of specificity for each gene-specific siRNA knockdown.
5. I wonder whether MUC5AC expression in response to HDM is independent on TRIF and whether phosphorylation of sNASP is TRIF-independent.
6. Figure 2, It would be better, the authors also add the data demonstrate dynamics of sNASP/TRAF6 interaction by IP for TRAF6 and blot for sNASP.
7. sFig. 2, There is no band in the sNASP-pSerine blot presented in supplemental figure 2. In addition, it seemed incorrect information is provided in supplement figure 2 legend. Clarify whether the cells were stimulated with HDM for 24 hr. Clarify whether sNASP-pSerine is detected by IB or IP with anti-sNASP followed by anti-pSerine blot.
8. It would be nice if the authors add the data showing whether HDM stimulation affects interaction between S158A mutant and TRAF6.
9. Figure 3, The better control treatment would be PEP-scramble peptide than NASP peptide.
10. Throughout the manuscript, term “transduction” usually indicates “the transfer of genetic material by a viral agent”. The process used in this study is adding cell-permeable peptides into the culture media. Thus, “treatment” might be the better word choice.
11. Page 11 “Our study demonstrated that HDM signifi-cantly induced the protein expression of MUC5AC, but not MUC5B (data not shown), which is involved in the activation of the TLR2/4, TRAF6, MAPK (p38), and NF-B signal-ing pathways.” Clarify this sentence. I assume what the authors meant that HDM induce expression of MUC5AC through activation of TLR2/4… and NF-kB signaling pathway not MUC5AC is involved in activation of TLR…NF-kB signaling pathway.
12. Page 12 “In our HDM-induced asthma model, …..reduced the production of ….. and Muc5ac.” It is about protein production. So, “Muc5ac” should be “MUC5AC”.
13. Page 13 “… and injected by intranasal 1 h before …”. Describe how to do intranasal injection.
Major concerns must be addressed:
1. Fig. 2A, This reviewer see neither phosphorylated sNASP bands nor actin bands. Check these blots presented in Fig. 2A and provide correct blots.
2. Page 3 &4, “At 30 minutes after HDM stimulation, sNASP was serine-phosphorylated in 16HBE cells (Figure 2A). Similar to our previous finding, endog-enous TRAF6 dissociation from sNASP was correlated with increased phosphorylation of sNASP in response to HDM (Figure 2A). These results suggested that HDM-treated bron-chial epithelial cells can initiate phosphorylation of sNASP, which leads to the release of TRAF6 from sNASP to allow downstream signaling”. These claim by the authors are not supported by the data presented in Fig. 2A. Either show the supporting data or delete this claim.
3. Figure 7, The authors must provide information on how they detect the phosphorylated form of sNASP in IHC and WB and how they validate what the presented data.
4. Page 13 “Phosphorylation of sNASP in the lung sections were detected by using anti–phospho-sNASP S158.” Provide source of anti-phospho-sNASP S158. If it is a custom antibody and reported previously, please cite the previous paper that showing validation of the antibody. If it has not been previously reported, please describe how the antibody is generated and validated.
Minor concerns need to be clarified:
1. Page 2 and throughout the manuscript, “… TLRs activation” should be “.. TLR activation”. or “…. activation of TLRs”.
2. Page 2 and throughout the manuscript, “…TLRs-mediated..” & “…TLRs signaling..” should be “.. TLR-mediated..” & “..TLR signaling..”, respectively.
3. Page 3 & page 4, “..at the level of both the mRNA and protein levels ….” remove “levels” following protein.
4. Supplementary information, it would be better if the authors provide evidence of specificity for each gene-specific siRNA knockdown.
5. I wonder whether MUC5AC expression in response to HDM is independent on TRIF and whether phosphorylation of sNASP is TRIF-independent.
6. Figure 2, It would be better, the authors also add the data demonstrate dynamics of sNASP/TRAF6 interaction by IP for TRAF6 and blot for sNASP.
7. sFig. 2, There is no band in the sNASP-pSerine blot presented in supplemental figure 2. In addition, it seemed incorrect information is provided in supplement figure 2 legend. Clarify whether the cells were stimulated with HDM for 24 hr. Clarify whether sNASP-pSerine is detected by IB or IP with anti-sNASP followed by anti-pSerine blot.
8. It would be nice if the authors add the data showing whether HDM stimulation affects interaction between S158A mutant and TRAF6.
9. Figure 3, The better control treatment would be PEP-scramble peptide than NASP peptide.
10. Throughout the manuscript, term “transduction” usually indicates “the transfer of genetic material by a viral agent”. The process used in this study is adding cell-permeable peptides into the culture media. Thus, “treatment” might be the better word choice.
11. Page 11 “Our study demonstrated that HDM signifi-cantly induced the protein expression of MUC5AC, but not MUC5B (data not shown), which is involved in the activation of the TLR2/4, TRAF6, MAPK (p38), and NF-B signal-ing pathways.” Clarify this sentence. I assume what the authors meant that HDM induce expression of MUC5AC through activation of TLR2/4… and NF-kB signaling pathway not MUC5AC is involved in activation of TLR…NF-kB signaling pathway.
12. Page 12 “In our HDM-induced asthma model, …..reduced the production of ….. and Muc5ac.” It is about protein production. So, “Muc5ac” should be “MUC5AC”.
13. Page 13 “… and injected by intranasal 1 h before …”. Describe how to do intranasal injection.
